

# Reduced P2X receptor levels are associated with antidepressant effect in the learned helplessness model

Deidiane Elisa Ribeiro[1,2,3,*], Plinio C. Casarotto[4,*], Laura Staquini[1], Maria Augusta Pinto e Silva[5], Caroline Biojone[4], Gregers Wegener[2] and Samia Joca[5,6]

[1] Department of Pharmacology, School of Medicine of Ribeirão Preto, University of São Paulo, Ribeirão Preto, Brazil
[2] Department of Clinical Medicine, Translational Neuropsychiatry Unit, Aarhus University, Aarhus, Denmark
[3] Department of Biochemistry, Chemistry Institute, University of São Paulo, São Paulo, Brazil
[4] Neuroscience Center—HILIFE, University of Helsinki, Helsinki, Finland
[5] Department of Physics and Chemistry, School of Pharmaceutical Sciences of Ribeirão Preto, University of São Paulo, Ribeirão Preto, Brazil
[6] Aarhus Institute of Advanced Studies (AIAS), Aarhus University, Aarhus, Denmark
* These authors contributed equally to this work.

Corresponding author
Samia Joca, samia@usp.br

## ABSTRACT

Purinergic receptors, especially P2RX, are associated to the severity of symptoms in patients suffering from depressive and bipolar disorders, and genetic deletion or pharmacological blockade of P2RX7 induces antidepressant-like effect in preclinical models. However, there is scarce evidence about the alterations in P2RX7 or P2RX4 levels and in behavioral consequences induced by previous exposure to stress, a major risk factor for depression in humans. In the present study, we evaluated the effect of imipramine (IMI) on P2RX7 and P2RX4 levels in dorsal and ventral hippocampus as well as in the frontal cortex of rats submitted to the pretest session of learned helplessness (LH) paradigm. Repeated, but not acute administration of IMI (15 mg/kg ip) reduced the levels of both P2RX7 and P2RX4 in the ventral, but not in dorsal hippocampus or frontal cortex. In addition, we tested the effect of P2RX7/ P2RX4 antagonist brilliant blue G (BBG: 25 or 50 mg/kg ip) on the LH paradigm. We observed that repeated (7 days) but not acute (1 day) treatment with BBG (50 mg) reduced the number of failures to escape the shocks in the test session, a parameter mimicked by the same regimen of IMI treatment. Taken together, our data indicates that pharmacological blockade or decrease in the expression of P2RX7 is associated to the antidepressant-like behavior observed in the LH paradigm after repeated drug administration.

## INTRODUCTION

Adenosine triphosphate (ATP)-mediated signaling has been recently involved in the behavioral effects of stress and neurobiology of depression (*Burnstock et al., 2011*; *Sperlagh et al., 2012*). ATP effects are mediated by the activation of P2 receptors (P2R), classified in

two major families: P2RX are ligand-gated ion channels, while P2RY are G protein-coupled receptors. Among those, P2RX have been associated to several processes that are dysfunctional in stress response and depression neurobiology, such as neurotransmitter release, cognition, sleep, energy levels, appetite, immune and endocrine system (*Burnstock et al., 2011*).

Accordingly, clinical evidence associates a P2RX7 polymorphism that results in a mutation in the protein (Q460R), with higher severity of symptoms in patients with major depressive disorder (MDD) (*Hejjas et al., 2009*), which was recently confirmed by a meta-analysis study (*Czamara, Müller-Myhsok & Lucae, 2018*).

Pre-clinical studies indicate that knocking out of P2RX7 leads to antidepressant-like phenotype in the forced swimming (FST) and tail suspension (TST) tests (*Basso et al., 2009*; *Boucher et al., 2011*; *Csölle et al., 2013a*, *2013b*). Corroborating this idea, systemic treatment with P2RX7 antagonist induces antidepressant-like effects in both FST and TST (*Csölle et al., 2013a*; *Pereira et al., 2013*). However, these two tests do not involve a previous exposure to stress, a major factor in triggering depressive behavior (*Hammen, 2005*). Indeed, a positive correlation between previous exposure to stressful events and the severity of the first depressive episodes has been demonstrated (*Post, 1992*). Two studies monitoring 1,942 adult women during 9 years (*Kendler, Gardner & Prescott, 2002*), and 2,935 adult men during 2–4 years (*Kendler, Gardner & Prescott, 2006*) elaborated a model to predict depressive episodes based on the patient's history. According to these findings, MDD would be the result of the interaction between risk factors from multiple domains, including stress exposure (*Kendler, Gardner & Prescott, 2002*, *2006*).

In this scenario, animal models that address the behavioral consequences of stress rise as useful tools to study the neurobiology of depression as well as to investigate potential new antidepressant drugs. One of the most prominent, the learned helplessness (LH) paradigm, presents good face validity since the exposure to inescapable foot shocks leads to endocrine, neuroanatomical and neurochemical changes observed in depression such as decreased hippocampal volume, diminished neurogenesis, impaired monoaminergic neurotransmission and hypothalamic pituitary adrenal axis imbalance (*Pryce et al., 2011*). In addition, helplessness has been found in depressed patients which turned it into the focus of preclinical and clinical depression research (*Seligman, 1975*; *Pryce et al., 2011*). The LH predictive validity is supported by the lack of responsiveness to acute treatment with classical antidepressants, as observed in FST (*Saarelainen et al., 2003*), but it requires about 7 days of continuous treatment to induce observable effects (*Petty, Kramer & Wilson, 1992*). Moreover, LH is not sensitive to anxiolytic drugs (diazepam, lorazepam) or to stimulants (amphetamine, caffeine), which further supports that the model exhibits good predictive validity (*Sherman, Sacquitne & Petty, 1982*; *Martin & Puech, 1996*).

Despite previous evidence indicating that P2RX7 blockade might induce antidepressant-like effect, it is not clear if P2RX7 antagonists are effective in the LH model. Moreover, it has not yet been investigated if repeated treatment with these drugs is required to induce antidepressant-like effects. Thus, the aim of the present study was to: (1) evaluate the effect of acute and repeated treatment with the P2RX7 antagonist brilliant blue G (BBG) in animals submitted to the LH model; (2) to investigate if the stress and the

antidepressant effect would be associated with altered P2RX7 levels in the hippocampus and frontal cortex, brain regions highly implicated in the neurobiology of depression (*Liu et al., 2017*). In addition, P2RX4 levels were also evaluated since these receptors may form heterodimers with P2RX7 (*Guo et al., 2007*), P2RX4 activation modulates depressive-related behaviors (*Bortolato et al., 2013*) and BBG may also block P2RX4 (*Jiang et al., 2000*).

## METHODS

### Animals

Male Wistar rats weighting 250–280 grams (about 8 weeks old) were obtained from the Central Vivarium of the University of São Paulo, campus of Ribeirão Preto, Brazil. The animals were kept under standard conditions: temperature (24 ± 1 °C), light cycle (lights on from 6:00 a.m. to 6:00 p.m.), free access to food and water. The animals were kept in groups of 4/cage (41 × 34 × 16 cm) during the habituation period (at least 1 week prior to the beginning of experiments) and isolated (30 × 20 × 13 cm) during the experimental procedure. The welfare of the animals was assessed daily. The cages and bedding were changed every 2 days as well as food and water replacement. Animals were randomly assigned to the different experimental groups and experiments were conducted from 7 am to 6 pm, with randomization of treatment conditions. All procedures were developed in accordance with Brazilian Council for Animal Experimentation (CONCEA), they were approved by the ethical committee for animals use of the School of Pharmaceutical Sciences of Ribeirão Preto, University of São Paulo (CEUA, Protocol Number 13.1.1506.53.0), and all efforts were made to minimize animal suffering.

Based on previous experiments, the sample size required for LH experiments is about 10 animals per group (*Stanquini et al., 2017*). In the present study, the total number of animals used was 126 male rats, an average of 8.4 animals/group.

### Drugs and reagents

The following drugs were freshly prepared before use and administered intraperitoneally (ip): BBG (#B0770; Sigma Aldrich, St. Louis, MO, USA), a P2RX7/P2RX4 antagonist (*Jiang et al., 2000*), diluted in sterile isotonic saline and 2% tween 80, administered at 25 or 50 mg/kg/day doses (*Csölle et al., 2013a*; *Carmo et al., 2014*). Imipramine (IMI; #I7379; Sigma Aldrich, St. Louis, MO, USA), a tricyclic antidepressant, diluted in sterile isotonic saline, administered at 15 mg/kg/day dose (*Stanquini et al., 2017*). Chloral hydrate (Vetec) was used as a sedative for sample collection, at 5% concentration and administered at one ml/100g volume.

### Learned helplessness

The apparatus consisted of an acrylic box (30.7 × 33.3 × 54 cm, INSIGHT Apparatus— EP111, Brazil), with two compartments of equal sizes separated by a wall with a central opening, through which the animals could cross from one compartment to another. The apparatus has a metal grid on the floor, which can deliver foot shocks. The behavioral tests took place in a sound attenuated, temperature-controlled (25 ± 1 °C) room.

The experimental procedure was conducted similarly to previously described (*Stanquini et al., 2017*). Male Wistar rats were exposed to a pre-test (PT) session, consisting of 40 inescapable foot shocks (0.4 mA, 10 s duration, 30–90 s interval). 7 days later, the rats were submitted to the test session (T), when 30 escapable foot shocks (0.4 mA, 10 s, 30–90 s interval), preceded (5s) by a warning tone (60dB, 670Hz) were applied. In the test session, the animals could avoid (by crossing from one compartment to the other during tone presentation) or escape (by interrupting the shock when crossing the shuttle box during shock application) from the shocks. Inescapable stress exposure in PT leads to helplessness behavior, reflected as failure to avoid/escape the shocks in the test session. Antidepressant treatment decreases the number of failures to avoid/escape shocks (*Sherman, Sacquitne & Petty, 1982*). Therefore, the number of escape failures was registered in the present work as a parameter indicative of LH behavior. The number of inter trial crossings (ITC) were also recorded as a parameter of locomotor activity (*Geoffroy & Christensen, 1993*).

## Sample collection, preparation and western blotting analysis

The animals were deeply anesthetized with chloral hydrate 5% and decapitated. The frontal cortex, ventral and dorsal hippocampus were dissected on ice for posterior analysis. Considering the longitudinal axis of the rat brain, dorsal and ventral hippocampus were respectively divided in the two-thirds higher and one-third lower of the total hippocampus due their differences in anatomical connections, patterns of gene expression and behavioral functionality (*Bannerman et al., 2014*; *Strange et al., 2014*). Samples were mechanically homogenized in lysis buffer (137 mM NaCl, 20 mM Tris–HCl pH 7.6, 10% glycerol), supplemented with protease inhibitor cocktail (#P2714; Sigma Aldrich, St. Louis, MO, USA). The homogenate was centrifuged for 15 min, 9,000×*g*, 4 °C. The supernatant was collected and stored at −80 °C.

The levels of P2RX7 and P2RX4 were determined in frontal cortex, ventral and dorsal hippocampus samples by western blotting (WB). Briefly, quantification of total proteins was used to determine the amount of each sample was submitted to analysis (*Bradford, 1976*). Then, 30 micrograms of proteins from each sample were resolved in SDS-PAGE (12% polyacrylamide gel) and transferred to a polyvinylidene fluoride membrane. The membranes were blocked with 5% bovine serum albumin solution in tris buffered saline with tween 20 (TBST) buffer and incubated with primary antibody against P2RX7 (1:200, #APR-004; Alomone Labs, Jerusalem, Israel), P2RX4 (1:500, #APR-002; Alomone Labs, Jerusalem, Israel) or GAPDH (1:1,000, #sc-25778; Santa Cruz, Dallas, TX, USA), overnight, 4 °C.

The membranes were incubated with horseradish peroxidase (HRP)-conjugated secondary antibody (goat anti-rabbit, 1:2,000, #7074; Cell Signaling, Danvers, MA, USA) for 1 h at room temperature, and washed with TBST buffer. The HRP activity was developed with chromogenic reagent 4-chloro-naphthol (#NEL300001EA; Perkin Elmer, Waltham, MA, USA). Finally, the membranes were air dried and scanned, the optical density was analyzed with the Image Studio Lite program (version 5.2) and the values for the P2RX7 or P2RX4 were normalized by the GAPDH value in the corresponding sample and expressed as a percentage of the control group (vehicle-injected).

## Experimental design

The animals assigned to WB analysis were maintained in the animal house for at least 1 week before the beginning of the experimental protocol. Animals were brought to the lab on day 1 and taken individually to experimental rooms, where they were exposed to PT. Right after, animals received the administration of IMI or vehicle. After that, they were housed in individual cages (30 × 19 × 13 cm) and taken back to the animal house. Every day, at 11:00 am, the animals were brought to the lab experimental rooms, where they were weighted and injected with drug or vehicle, according to the groups they had been randomly assigned to. Acutely treated animals received the administration of vehicle during 6 days and, in the 7th day, they were injected with IMI. On the last day, animals received the last injection and returned to their homecages where they remained undisturbed for 1 h. Following, they were anesthetized and euthanized for frontal cortex, dorsal and ventral hippocampus dissection. The last treatment injection and sample collection were performed in random order to avoid circadian influences on the analysis.

The animals submitted to LH paradigm were kept in groups of four/cage (41 × 34 × 16 cm) for at least 1 week before the beginning of the experiment. For behavioral experiments, animals were brought to the lab on day 1 and taken individually to experimental rooms, where they were exposed to PT. Immediately after completion of PT, animals received the administration of drug (BBG or IMI) or vehicle. After that, they were housed in individual cages (30 × 19 × 13 cm) and taken back to the animal house. Every day, at 11:00 am, the animals were brought to the lab experimental rooms, where they were weighed and injected with drug or vehicle, according to the groups they had been randomly assigned to. Acutely treated animals received the administration of vehicle during 6 days and, on the 7th day, they were injected with drug (BBG or IMI). 1 h after the last injection, the animals were exposed to the T session of LH paradigm.

## Statistical analysis

The number of escape failures and ITC in the LH were analyzed by Kruskal–Wallis test followed by Dunn's post hoc. Data from WB was analyzed by one-way ANOVA followed by Fisher's post hoc test. Values of $p < 0.05$ were considered statistically significant. All data used in the present study is available in FigShare under CC-BY license (DOI 10.6084/m9.figshare.6989063).

## RESULTS

### Experiment 1: effect of imipramine in the P2RX7 and P2RX4 levels of rats exposed to LH model

One-way ANOVA indicated a significant effect of IMI treatment on P2XR7 ($F(2,18) = 4.169$, $p = 0.0325$) and P2RX4 ($F(2,20) = 3.627$, $p = 0.0453$) levels in ventral hippocampus. Fisher's test showed that repeated IMI-treatment significantly reduced the level of both receptors in comparison to vehicle-treated groups ($p < 0.05$), as found in Fig. 1A.

In samples from dorsal hippocampus (Fig. 1B) or frontal cortex (Fig. 1C), no effect of treatment with IMI was observed.

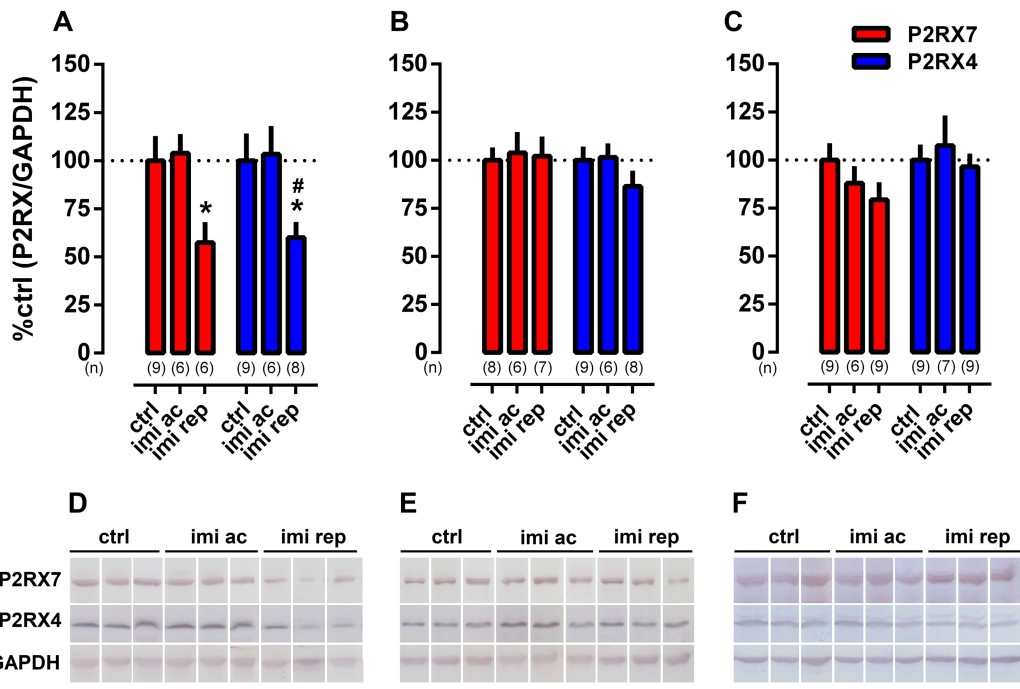

**Figure 1 Effect of repeated treatment with imipramine on the levels of P2RX7 and P2RX4 in the (A) ventral hippocampus, (B) dorsal hippocampus and (C) frontal cortex.** Representative WB bands of (D) ventral hippocampus, (E) dorsal hippocampus and (F) frontal cortex. Values are expressed as mean ± SEM and the sample size of each experimental group ($n$) is depicted under the respective columns. *$p < 0.05$ from ctrl (vehicle-treated) group; #$p < 0.05$ from imi ac group.

## Experiment 2: effect of BBG and imipramine treatment in the LH paradigm

Kruskal–Wallis test showed that acute treatment with BBG altered neither the number of failures ($H = 1.75$, $p = 0.42$) nor the ITC ($H = 4.52$, $p = 0.10$) of animals exposed to the LH model (Fig. 2A). However, repeated treatment with BBG decreased the number of escape failures ($H = 10.53$, $p = 0.0052$) without changing the ITC ($H = 0.27$, $p = 0.87$) of animals exposed to the LH model treated with BBG 50mg/kg (Dunn's $p < 0.05$) (Fig. 2B).

Repeated but not acute treatment with IMI (15 mg/kg) significantly decreased the number of escape failures ($H = 7.949$, $p = 0.019$) and induced no alterations in the ITC ($H = 1.41$, $p = 0.49$) of animals exposed to the LH paradigm (Fig. 2C).

## DISCUSSION

In the present study, we report that the antidepressant-like effect of IMI is associated with the attenuation of P2RX7 and P2RX4 levels in ventral hippocampus of rats exposed to LH. Repeated but not acute administration of IMI, an antidepressant consistently tested in the LH model (*Sherman, Sacquitne & Petty, 1982*; *Takamori, Yoshida & Okuyama, 2001*; *Joca, Padovan & Guimarães, 2003*; *Stanquini et al., 2017*), decreased the number of escape failures of animals exposed to this paradigm, as well as P2RX7 and P2RX4 levels in the ventral hippocampus of stressed animals.

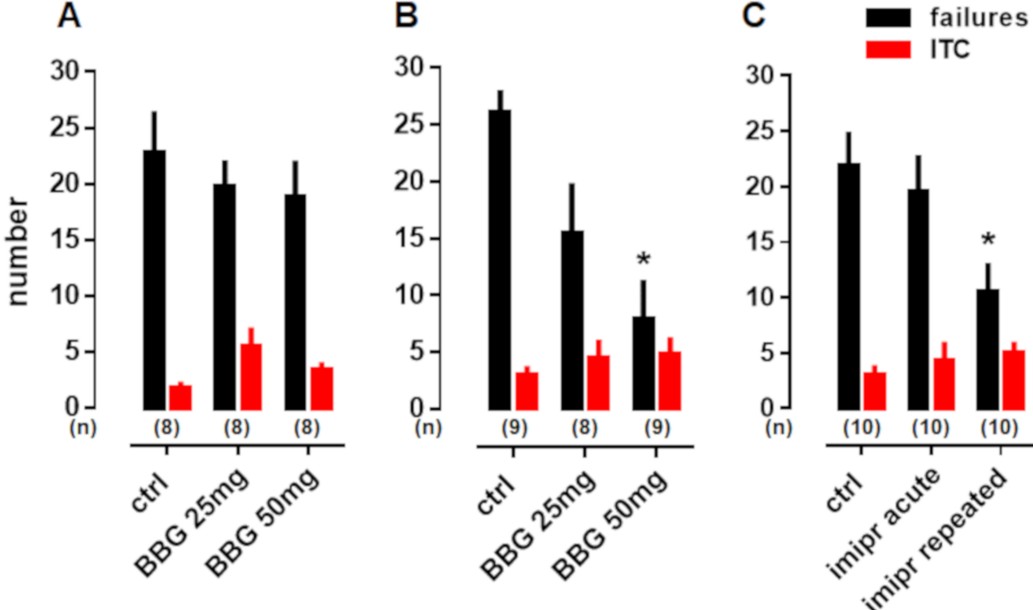

**Figure 2 Effect of BBG and imipramine treatment in the LH paradigm.** (A) Acute treatment with BBG did not alter the number of failures (black) or the ITC (red) of animals exposed to the LH paradigm. (B) Repeated treatment with BBG 50 mg/kg decreased the number of failures without changing the ITC of rats submitted to this model. (C) Repeated but not acute administration of imipramine decreased the number of failures in the LH paradigm without altering the ITC. Values are expressed as mean ± SEM and the sample size of each experimental group (n) is depicted above the respective column. *$p < 0.05$ from ctrl (vehicle-treated) group.

Based on the findings described above, we further assessed if acute and repeated treatment with P2RX7/P2RX4 antagonist, BBG, could induce antidepressant effects in animals exposed to the LH model. By doing that, we assessed if repeated treatment with BBG would be required to induce antidepressant effects. The LH model was chosen based on its fulfillment of face, construct and predictive validity criteria (*Sherman, Sacquitne & Petty, 1982*; *Willner, 1984*; *Maier et al., 1986*; *Petty, Kramer & Wilson, 1992*; *Nankai et al., 1995*; *Centeno & Volosin, 1997*; *Fleshner et al., 1998*; *Maier & Watkins, 2005*; *Hajszan et al., 2009*). Similar to the clinical scenario, and contrary to usual protocols of FST and TST, the antidepressant-like effect on LH model is observed only after repeated pharmacological treatment, i.e., administration of classical antidepressant drugs during 7–21 days (*Petty, Kramer & Wilson, 1992*; *Nankai et al., 1995*; *Mansbach, Brooks & Chen, 1997*; *Hajszan et al., 2009*). We observed that only the repeated (7 days) treatment with BBG decreased the number of escape failures. None of the treatments changed the number of ITC performed by animals exposed to the LH. The ITC was evaluated as a proxy of the locomotor activity since alterations in this parameter can interfere in the animals' behavior during the LH test (*Geoffroy & Christensen, 1993*). Altogether, these results are interpreted as an antidepressant-like effect induced by BBG (*Sherman, Sacquitne & Petty, 1982*).

Previous studies indicated that BBG exerts antidepressant-like effect in animals submitted to the FST and TST (*Csölle et al., 2013a*). Chronic treatment with A-804598, a selective P2RX7 antagonist, was able to block the anhedonic effects of chronic

unpredictable stress (*Iwata et al., 2016*). Moreover, mice lacking P2RX7 showed decreased LH behavior (*Otrokocsi, Kittel & Sperlágh, 2017*). Our results further support these findings and add important information regarding the antidepressant effect induced by P2RX7 blockade, since none of the previous studies assessed the effects induced by the acute treatment. In light of that, our findings suggest that repeated treatment with BBG is required to induce antidepressant effects, similarly to conventional monoaminergic antidepressants.

In fact, the antidepressant effect induced by P2RX antagonists is dependent on the facilitatory action upon monoamine signaling in the central nervous system (CNS) since depletion of serotonin or noradrenaline blocked the effects induced by pyridoxalphosphate-6-azophenyl-2,4-disulphonoic acid (*Diniz et al., 2017*). Moreover, chronic blockade of P2RX7 have also shown to increase brain-derived neurotrophic factor (BDNF) levels in mice hippocampus (*Csölle et al., 2013a*) as well as activate BDNF-TRKB signaling pathway in ventral hippocampus of stressed rats (*Ribeiro et al., 2019*), an effect that is central to the mechanism of action induced by monoaminergic drugs (*Saarelainen et al., 2003*; *Rantamäki et al., 2007*). Therefore, it is plausible to suggest that P2RX7 blockade could promote antidepressant effects as a result of monoaminergic and BDNF signaling facilitation.

In addition, activation of the tropomyosin receptor kinase B (TRKB) receptor-mediated signaling by antidepressants could partially counteract the consequences of P2RX7 activity, putatively through epigenetic mechanisms (*Duclot & Kabbaj, 2015*), such as chemical modifications in the DNA or histones (*Vialou et al., 2013*; *Nestler, 2014*). These mechanisms are also found to regulate P2RX7 gene transcription (*Zhou et al., 2009*; *Shin et al., 2015*). Therefore, the activation of TRKB receptors would result in an epigenetic-mediated decrease in transcription and, consequently, expression of P2RX7, contributing to the antidepressant-like effect.

P2RX7 are cation channels activated by high concentrations of ATP, with an EC50 around one mM (*Donnelly-Roberts et al., 2009*), which is released after stress exposure (*Volonte et al., 2012*; *Jiang et al., 2013*). P2RX7 are widely expressed in the CNS including brain regions involved in stress response such as frontal cortex and hippocampus (*Jimenez-Mateos et al., 2018*). For this reason, we focused our molecular analysis in these two brain regions. These receptors were assumed to be expressed in nerve terminals but an ongoing debate suggests that the observed effects would be an indirect result of P2RX7 activation in glial cells (for a detailed discussion see (*Illes, Khan & Rubini, 2017*)).

P2RX7 stimulation leads to the activation of neuronal nitric oxide synthase (NOS1) (*Pereira et al., 2013*), which in turn provides a positive feedback to glutamate release. P2RX7 activation also promotes potassium efflux, thus stimulating the nucleotide-binding, leucine-rich repeat, pyrin domain containing 3 (NLPR3) inflammasome and caspase 1 (CASP1), leading to interleukin release (mainly IL-1β and IL-6). Therefore, enhancing neuroimmune response, axonal degeneration, cell death and inhibiting neurogenesis through NFkB signaling (*Iwata et al., 2016*). The whole process could be positively fed by the decrease in BDNF signaling promoted by P2RX7 activation (*Csölle et al., 2013a*).

Therefore, is plausible to suppose that the behavioral effects of antidepressants potentially involve the inhibition/repression of P2RX7.

P2RX4 are cation channels activated by ATP and also widely expressed in the CNS, including neurons and glial cells (*Soto et al., 1996*; *Stokes et al., 2017*). According to electron microscopy evidence, this receptor can be found on post- or pre-synaptic terminals (*Rubio & Soto, 2001*), and high levels of P2rx4 mRNA was detected in the dentate gyrus' granule cells and in CA1/CA3 pyramidal neurons as well as astrocytes in rat hippocampus (*Soto et al., 1996*; *Kukley et al., 2001*). Hippocampal P2RX4 have been associated to induction of NMDA-dependent long-term potentiation (LTP) (*Sim et al., 2006*; *Choi et al., 2010*). Severe stress decreases LTP in the rodent hippocampus, which has been associated with cognitive impairment (*Foy et al., 1987*; *Shors et al., 1989*; *Kim & Diamond, 2002*). However, stress also increases LTP (*Joels & Krugers, 2007*), which could be associated to the decrease in P2RX4 levels after antidepressant treatment. P2RX4 expressed in microglia (*Tsuda et al., 2003*; *Ulmann et al., 2008*) have been involved in the activation and migration of these cells to injury sites (*Guo, Trautmann & Schluesener, 2005*; *Schwab, Guo & Schluesener, 2005*).

In this context, stress might activate hippocampal P2RX4 increasing microglia/neuroimmune response (*Guo, Trautmann & Schluesener, 2005*; *Schwab, Guo & Schluesener, 2005*) as well as LTP and synaptic plasticity strengthening, contributing to the formation of aversive memories (*Sim et al., 2006*; *Baxter et al., 2011*). Such effects are frequently associated to depressive-related behaviors and can be prevented or reverted by antidepressant treatment (*Veith et al., 1994*; *Bliss & Cooke, 2011*; *Kreisel et al., 2014*). Accordingly, we observed that the repeated treatment with IMI decreased the P2RX4 expression in ventral hippocampus.

P2RX4 and P2RX7 are activated by ATP, which levels are efficiently controlled by an extracellular enzymatic chain collectively called of ectonucleoside triphosphate diphosphohydrolases (E-NTPDases) (*Yegutkin, 2014*). The treatment with three different doses (100, 250 or 300 μM) of antidepressants (fluoxetine, sertraline, nortriptyline or clomipramine) decreased E-NTPDases activity in hippocampal and cortical synaptosomes of rats (*Pedrazza et al., 2007, 2008*).

Ectonucleoside triphosphate diphosphohydrolase 1, also known as cluster of differentiation 39, is the rate-limiting enzyme of a cascade which contributes to extracellular adenosine production through the hydrolysis of ATP/ADP to AMP (*Yegutkin, 2014*) and antidepressant treatment seems to modulates this enzyme activity. Chronic treatment (10 mg/kg/day i.p. during 14 days) with nortriptyline promoted a decrease in NTPDase1 transcript levels in the hippocampus and induced an increase of gene expression for NTPDase1 in cerebral cortex while the same treatment regimen with fluoxetine produced an enhancement for NTPDase1 transcript levels in hippocampus and cerebral cortex of rats (*Pedrazza et al., 2008*).

These results indicate that antidepressants decrease E-NTPDases activity in the cortex and hippocampus which probably leads to increased levels of ATP. Although at first sight these results are contrary to our hypothesis, E-NTPDases activity measurements do not predict the ATP action upon P2RX4 or P2RX7.

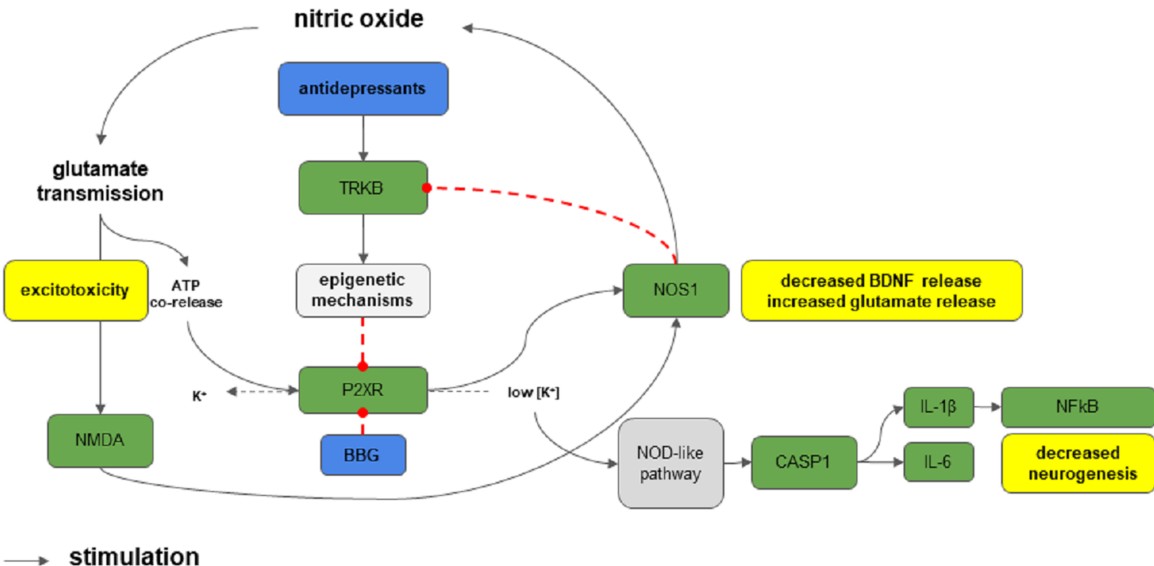

**Figure 3  P2RX7 signaling in stress.** Stress elicits massive glutamate and ATP release. The activation of P2RX7 receptors by ATP activates nitric oxide (NO) production, which provides a positive feedback to glutamate release. This process leads to excitotoxicity. P2RX7 activation also promotes potassium efflux, stimulating the nucleotide-binding, leucine-rich repeat, pyrin domain containing 3 (NLPR3) inflammasome and caspase 1 (CASP1), production of interleukins (IL-6 and IL-1β), and activation of NF-κB exacerbating neuroimmune response, axonal degeneration and cell death, and inhibition of neurogenesis. The whole process is positively fed by the decrease in BDNF signaling promoted by P2RX7. Antidepressants, putatively through TRKB-dependent epigenetic mechanisms could counteract such effects by decreasing P2RX7 expression. Green boxes: proteins or targets involved; blue: drugs or compounds; gray: process or pathways: red: deleterious effects.

## CONCLUSION

As shown in Fig. 3, excessive/sustained release of ATP during stress exposure could trigger the activation of P2RX7 and/or P2RX4 in brain regions important for stress and depression. In turn, this facilitates neurochemical and molecular processes that hinders behavioral adaptation to stress (i.e., increased inflammation and cell death through caspase), thus leading to behavioral consequences, such as LH and depression (*Burnstock et al., 2011*; *Sperlagh et al., 2012*). The increased nitric oxide production, by activation of P2RX7 or NMDA for example, could blunt TRKB activation, through a decrease of BDNF release (*Canossa et al., 2002*) or putative nitration of the receptor (*Biojone et al., 2015*).

In this scenario, attenuation of the P2RX7/P2RX4 levels by antidepressant could be part of the mechanisms that contribute to their behavioral/therapeutic effects and the selective blockade of these receptors could represent a new strategy to develop novel antidepressant drugs. Although our data supports this hypothesis, further studies are required to elucidate the P2RX7 and P2RX4 role in stress response. Altogether our data shows that inhibition of P2RX4- and P2RX7-mediated signaling by BBG or IMI induces antidepressant-like effects.

## ACKNOWLEDGEMENTS

Authors are thankful for the technical assistance of Flavia Salata (University of São Paulo).

### Funding

This study was funded by the State of São Paulo Research Foundation (2013/01737-7) and Aarhus University Research Foundation (AU-UDEAS initiative: eMOOD, and a Mobility Stipend to Deidiane Elisa Ribeiro). The funders had no role in study design, data collection and analysis, decision to publish, or preparation of the manuscript.

### Grant Disclosures

The following grant information was disclosed by the authors:
State of São Paulo Research Foundation: 2013/01737-7.
Aarhus University Research Foundation (AU-UDEAS initiative: eMOOD, and a Mobility Stipend).

### Competing Interests

Gregers Wegener received lecture/consultancy fees from H. Lundbeck A/S, Servier SA, Astra Zeneca AB, Eli Lilly A/S, Sun Pharma Pty Ltd, Pfizer Inc, Shire A/S, HB Pharma A/S, Arla Foods A.m.b.A., Alkermes Inc, and Mundipharma International Ltd. All other authors declare no conflict of interest.

### Author Contributions

- Deidiane Elisa Ribeiro conceived and designed the experiments, performed the experiments, analyzed the data, prepared figures and/or tables, authored or reviewed drafts of the paper, approved the final draft.
- Plinio C. Casarotto conceived and designed the experiments, performed the experiments, analyzed the data, prepared figures and/or tables, authored or reviewed drafts of the paper, approved the final draft.
- Laura Staquini performed the experiments, analyzed the data, prepared figures and/or tables, authored or reviewed drafts of the paper, approved the final draft.
- Maria Augusta Pinto e Silva performed the experiments, analyzed the data, prepared figures and/or tables, authored or reviewed drafts of the paper, approved the final draft.
- Caroline Biojone conceived and designed the experiments, performed the experiments, analyzed the data, authored or reviewed drafts of the paper, approved the final draft.
- Gregers Wegener conceived and designed the experiments, analyzed the data, contributed reagents/materials/analysis tools, authored or reviewed drafts of the paper, approved the final draft.
- Samia Joca conceived and designed the experiments, analyzed the data, contributed reagents/materials/analysis tools, prepared figures and/or tables, authored or reviewed drafts of the paper, approved the final draft.

## Animal Ethics

The following information was supplied relating to ethical approvals (i.e., approving body and any reference numbers):

All procedures were developed in accordance with Brazilian Council for Animal Experimentation (CONCEA), they were approved by the ethical committee for animals use of the School of Pharmaceutical Sciences of Ribeirão Preto, University of São Paulo (CEUA, Protocol Number 13.1.1506.53.0), and all efforts were made to minimize animal suffering.

## Data Availability

All data used in the present study is available in FigShare: Casarotto, Plinio (2019): data. figshare. Dataset. https://doi.org/10.6084/m9.figshare.6989063.v3.

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
