# Peer review of "Reduced P2X receptor levels are associated with antidepressant effect in the learned helplessness model"

_PeerJ, doi:10.7717/peerj.7834_

## Round 0.1 · original submission · Major Revisions

Based on the advice received, I have decided that your manuscript could be reconsidered for publication should you be prepared to incorporate major revisions suggested by the reviewers.

I would ask you to respond at all to the points raised by the reviewers. Please pay particular attention to data presentation (representative western blot images, divided the results into two figures, one referring to the WB and the other to the behavioral analysis, loading control of protein).

Reviewer 1 ·

Basic reporting

-The article was written in clear and professional English. In addition, this study presents literature references consistent. However, there is an excessive number of references, which makes the article very speculative. Therefore references should be reviewed, prioritizing the most current references.
-Althougth the figures are relevant to the content of the article, of sufficient resolution, and appropriately described and labeled, the reorganization of the figures would contribute to a better understanding of the results. I suggest that the figures of the behavioral results and the western blot analyzes be separated and placed in the order in which the experiments were performed. For example, figure 1 behavioral results and figure 2 western blot analyzes. In addition, representative western blot images should be added to the article, the lack of these data is an important limitation of this study.

Experimental design

-The experimental design of this study was well delineated to respond to the proposed objective.
-Given that the experimental groups present a different sample size, I suggest that the n of each group in each experiment should be clarified in the materials and methods or in the legends of the figures in order that this study has enough information to be reproducible by another researcher.

Validity of the findings

-The authors provided robust and statistically adequate data. A minor point is the lack of information about how the authors evaluated the normality of the data, since they used a non-parametric test.

-An important point is about the conclusion of study.
Although the results of this study can support the hypothesis created by the authors. The conclusion did not appropriately stated. Please provide a conclusion connected to objective of study and limited to those supported by the results.

Additional comments

-Although speculation is welcome in this jounal, the discussion of this study is too speculative. In particular, the fifth paragraph of the discussion (lines 233-250) that needs more connection to the data of the article

-The least important points:
1-Some abbreviations appear without being spelled in the first use, for example: HRP and PPADS. Please ensure that uncommon abbreviations were spelled out at first use.
2- Do not acknowledge funders in the Acknowledgments session, there is a separate Funding Statement for that.
3-The sentence ..."and the values for the P2RX7 were normalized by the GAPDH value in each sample.." should be modified by ..."and the values for the P2RX7 and P2RX4 were normalized by the GAPDH value in each sample.."

Reviewer 2 ·

Basic reporting

In the manuscript “Reduced P2X receptor levels are associated with antidepressant effect in the learned helplessness model” the authors have reported that the genetic deletion or pharmacological blockade of purinergic receptors P2RX7, induces antidepressant-like effect in preclinical models. Moreover, they observed that the repeated administration of imipramine reduced the levels of both P2RX7 and P2RX4 in the hippocampus.
The study is very interesting, well written and show good results. However, there are some points that could be improved to make it suitable for publication in PeerJ. Therefore, I have given some suggestions which could help the authors to do it.

1. The raw data of the manuscript were not presented, in accordance with the Data Sharing policy.

Experimental design

2. The results would be clearer if divided into two figures, one referring to the Western Blot (WB) analysis and the other to the behavioral analysis.

3. In the results of WB, please add the images of the membranes to the figures 1a, 1b and 1c. Moreover, would be appropriate carried out a loading control of protein i.e.: β-actin, GAPDH…


4. The authors cite that P2RX7 is activated by high concentrations of ATP, which would have been supposed to be high. It would be interesting to quantify ATP levels in the analyzed tissues.

5. Still regarding the purinergic system, what happens to the levels of the enzymes that degrade nucleotides? Changes in ATP levels could be result or cause changes in activity or expression of these enzymes. Please add data about this. Experimental analyzes of CD39 levels, for example, would improve the work.

6. What is the reason for the differences between the receptor levels in different places in the brain? Please add an explanation about this in the discussion.

Validity of the findings

no comment

Additional comments

7. In the text, the results could be presented in the order in which the figures appear, starting at "a" or the figures could be rearranged.

8. I think that creating a figure, illustrating the possible mechanism by which antidepressants effects occur would be a way to highlight the results.

9. In the discussion, line 239-241, there is a repeated sentence.

---

## Round 0.2 · accepted · Accept

I have received the reviews on your manuscript, based on the advice received and my own evaluation, I am happy to inform you that your manuscript could be accepted for publication in PeerJ in its present form.

Reviewer 1 ·

Basic reporting

No comment.

Experimental design

No comment.

Validity of the findings

No comment.

Additional comments

I consider that the quality of the study has been improved and my questions were considered, then I recommend publishing this manuscript.

Reviewer 2 ·

Basic reporting

The authors properly answered the questions raised and added some important information, improving the quality of the manuscript.

Experimental design

The methodology is adequate and meets the objectives.

Validity of the findings

This work is of general interest and contributes to increase knowledge about the role of antidepressant on attenuation of the P2RX7/P2RX4 levels and the possible mechanisms that contribute to their behavioral/therapeutic effects.